# Advances in the Regulation of Neural Function by Infrared Light

**DOI:** 10.3390/ijms25020928

**Published:** 2024-01-11

**Authors:** Lequan Song, Hui Wang, Ruiyun Peng

**Affiliations:** Beijing Institute of Radiation Medicine, Beijing 100850, China

**Keywords:** infrared light, regulation, nerve function, mechanism

## Abstract

In recent years, with the rapid development of optical technology, infrared light has been increasingly used in biomedical fields. Research has shown that infrared light could play roles in light stimulation and biological regulation. Infrared light has been used to regulate neural function due to its high spatial resolution, safety and neural sensitivity and has been considered a useful method to replace traditional neural regulation approaches. Infrared neuromodulation methods have been used for neural activation, central nervous system disorder treatment and cognitive enhancement. Research on the regulation of neural function by infrared light stimulation began only recently, and the underlying mechanism remains unclear. This article reviews the characteristics of infrared light, the advantages and disadvantages of infrared neuromodulation, its effects on improving individual health, and its mechanism. This article aims to provide a reference for future research on the use of infrared neural regulation to treat neuropsychological disorders.

## 1. Introduction

Electromagnetic waves are ubiquitous in many aspects of our lives, and have become a major source of environmental pollution, while providing us with convenience. Infrared (IR), also known as IR radiation, is an electromagnetic wave with a frequency between microwave and visible light which has been researched extensively since its discovery more than 200 years ago. Through continuous development and exploration, its applications in industrial, agricultural, military and other fields have become more mature. In the field of neuroscience, IR research and applications have been less prevalent. IR was initially used for functional brain imaging, and the correlations between brain regions related to cognitive activities were inferred by detecting changes in blood oxygen levels [1]. Research has shown that IR-based methods can be used to treat nervous system and mental disorders, promote skin wound healing, relieve pain, stimulate mesenchymal and cardiac stem cell proliferation, and treat ophthalmic disorders [2,3,4,5]. The use of IR-based methods to regulate neural function has become a popular research topic in recent years, and these methods have considerable advantages over traditional neural stimulation technologies such as electrical stimulation, magnetic stimulation, and optogenetics. At present, IR has become an important tool in neuroscience research and the treatment of central nervous system disorders. We searched the Pubmed, Web of Science and Springer databases for the years 2005–2023 using keywords: infrared light, neuron and regulation. This article reviews the advantages and disadvantages of the IR-based method, its role in regulating neural function and improving individual health and its action mechanism based on research in the literature.

## 2. Roles of IR light in Neural Regulation

IR rays, also known as IR radiation, are radiation with a frequency of 0.3–400 THz and a wavelength of 0.75–1000 μm in vacuum. According to the wavelength, IR rays can be divided into near-infrared (NIR), mid-infrared (MIR) and far-infrared (FIR) rays, as shown in Table 1 [6]. Terahertz (THz) waves, which were commonly used in previous studies, are in the FIR band, and the rotational and vibrational levels of biological macromolecules overlap with the levels of THz waves, suggesting that THz waves could affect the structure and function of biological macromolecules and nerve cells, thereby regulating neural behavior [7].

The atoms or molecules that make up an object are in a constant state of vibration and rotation. When a molecule transitions from a higher energy level state to a lower energy level state, energy is radiated outwards in the form of electromagnetic waves, producing IR rays. In the lower energy level state, electromagnetic radiation of a certain energy is absorbed. In general, if the composition and structure of the object do not change, its energy level difference does not change. Therefore, the wavelengths of the electromagnetic waves absorbed and emitted by the object are essentially identical [8,9].

Neuromodulation is a biomedical engineering technology that uses electricity, magnetism, light and other means to alter the nervous system through invasive or noninvasive approaches or to regulate the activity of neurons and their neural networks, ultimately inducing changes in specific brain functions. Neuromodulation technology has advanced considerably over the past 30 years, providing a new research tool for basic neuroscience research and a new interventional method for the clinical treatment of nervous system disorders. Moreover, in addition to the neuromodulation effects of physical factors, exogenous drugs and neuromodulators in their natural this state could also exert neuromodulation effects under certain conditions [10,11]. IR neural modulation (INM) is a neural regulation technology that has emerged in recent years. Compared with other modulation methods, such as electrical stimulation, magnetic stimulation, optogenetics and visual stimulation, INM has substantial advantages (Table 2) [12,13,14].

INM methods directly regulate neural function through IR rays. Research has shown that INM methods can be used to stimulate central and peripheral nerves, inducing neurons that generate excitatory or inhibitory potentials, thereby regulating neural function. Compared to other modulatory methods, INM has the following advantages.

First, the spatial resolution of INM is high, and the stimulation is limited to the point of irradiation. The position of the stimulation can be changed by simply redirecting the beam. In contrast to electrical stimulation, INM is not affected by current diffusion. In addition, in contrast with optogenetics, INM can be used directly with neural tissue without gene level modification. Moreover, INM is insensitive to external electromagnetic fields and can be used with magnetic resonance or electrophysiology techniques [25,26,27].

However, the potential risk of INM is that it may cause tissue thermal damage and has limited stimulation depth depending on the IR absorption characteristics of the tissue [28].

## 3. IR Stimulation Can Be Used to Regulate Neural Function and Improve Health

Previous studies have shown that under certain conditions, IR could act directly on different types of nerve tissue or cells to regulate their excitability. INM has been used as a new treatment for central nervous system disorders that affect neurobehavior. In addition, INM can also be used to improve emotional and cognitive function in healthy individuals.

### 3.1. IR Light Can Regulate Neural Activity

IR neural activation, or IR stimulation of nerve tissue or neurons to generate action potentials, has been applied in several in vivo and in vitro models.

#### 3.1.1. Stimulating Effect on the Central Nervous System

Previous research on central nerve activation via INM has mainly focused on the NIR and MIR bands. INM could induce action potentials in the cortex and cochlear nerve. Previous research on the activation of the cortical nerve has mainly focused on the brain slice model, and research on the cochlear nerve has mainly focused on in vivo animal models. NIR activation mainly leads to thermal effects, while MIR could activate the central nervous system under nonthermal conditions.

NIR methods have long been used to activate the central nervous system. Some researchers have used IR diode lasers (λ = 1875 nm, H = 0.01–0.55 J/cm^2^) to emit NIR light to stimulate the somatosensory cortices of rats and quantified the neural response through electrophysiological and optical imaging methods. In this study, craniotomy and durotomy were conducted to expose somatosensory cortex, allowing NIR to irradiate the sensory cortex. The results showed that the response in the sensory cortex induced by INM was similar to the tactile response induced by stimulating the forepaw and whisker, and the amplitude of the evoked signal increased with increasing IR repetition frequency and radiation exposure [29]. The study in healthy subjects confirmed the activation of the central nervous system via IR. Song et al. [30] used continuous NIR with a wavelength of 820 nm and a power density of 310 mW/cm^2^ to stimulate the left forehead of subjects. Evoked action potentials were generated in the motor cortex, suggesting that IR stimulation in this band could improve cortical excitability.

Gold nanorods (GNRs) can be used to enhance the cell stimulation effect. When the wavelength of the incident light matches the resonance frequency of the GNRs, the GNRs can effectively convert the absorbed light energy into heat energy [31]. Geng J. et al. [32] used NIR with a power of 0.5 mW and a wavelength of 800 nm to stimulate human neuroblastoma cells (SH-SY5Y) labeled with GNRs and used calcium ion indicators (Fluo-4, AM) to detect calcium signal changes. The results showed that the average 10 s NIR stimulation could achieve Ca^2+^ light activation, and the release rate of Ca^2+^ from GNR-labeled neurons was 6 fold that of unlabeled GNRs.

Traditional cochlear implants use electrical stimulation, which has low spatial resolution and might cause mechanical damage to the tissue. The spatial resolution of IR nerve stimulation is high, and there is no direct contact with the tissue, so the side effects are minimal. Tian L et al. [33] used NIR with wavelengths of 980 nm and 810 nm and intensities of 10–100 mJ/cm^2^ to stimulate the cochlear nerves of deaf guinea pigs. The IR method with two wavelengths could stimulate the cochlea to produce auditory evoked potentials. IR with a wavelength of 980 nm produced a higher amplitude and lower radiation threshold. The 980 nm laser is superior to the 810 nm laser in terms of the stimulation efficiency. Izzo AD et al. [34] used NIR with a wavelength of 2020 nm and an intensity of 0.01–0.1 J/cm^2^ to stimulate the gerbil cochlea. Light stimulation induced the cochlea of normal and deaf animals to generate compound nerve action potentials (CNAPs). After long-term stimulation, the response amplitude of the cochlea to IR radiation remained essentially stable. One potential clinical application of this technology is the cochlear implant, which is expected to be used in the treatment and research of otological disorders.

In recent years, many studies have investigated the central nervous system stimulatory effects of MIR. Liu X et al. [35] irradiated mouse cortical brain slices with 5.6 μm and 0.003 μW/μm^2^ of MIR, and the temperature was controlled within a safe range during irradiation(ΔT = 0.52 ± 0.79 °C). Electrophysiological recordings showed that the K^+^ current increased significantly after irradiation, but the Na^+^ current did not change significantly. A mathematical model was established to simulate the dynamics of the K^+^ and Na^+^ channels, showing that the carbonyl group (−C=O) in the functional domain of the K^+^ channels, which was involved in ion selection, resonantly absorbed the MIR photon energy, improving the K^+^ permeability. In addition, the increased K^+^ current narrowed the action potential waveform of the neurons and produced shunt effects; as a result, the cells had weaker responses to weak stimuli and stronger responses to strong stimuli. Zhang J et al. [36] irradiated the mouse cortex with noninvasive MIR at the same wavelength and a power of 9 ± 0.5 mW. Two-photon calcium imaging, c-Fos staining, and the electrophysiological results showed that many cortical neurons were activated, and the degree of activation was proportional to the irradiation time. Thus, MIR at a wavelength of 5.6 μm could be used to regulate neural signals in a nonthermal manner. Cayce J M et al. [37] stimulated thalamocortical brain slices at a wavelength of 2.51–5.3 μm and an intensity of 0.19 J/cm^2^, and patch-clamp recorded action potentials were generated by thalamic neurons. The neural electrical activity evoked by IR was correlated with the wavelength and frequency of the laser. The threshold radiation for action potential generation decreased with increasing IR wavelength and repetition rate.

#### 3.1.2. Stimulation of Peripheral Nerves

Most studies on INM activation of peripheral nerves were performed in the NIR band. At present, action potentials evoked by INM have been recorded in the sciatic nerve, cavernous nerve, and other nerves. Cury J et al. [38] used NIR with a wavelength of 1470 nm and a radiation intensity in the range of 1.59–4.78 mJ/cm^2^ to stimulate the sciatic nerves of rats in vitro and recorded the CNAP. The threshold radiation dose range was 2.48–3.84 mJ/cm^2^. Based on the results, the authors suggested that the longer the IR wavelength was, the lower the water absorption coefficient and the lower the energy threshold were; however, the threshold level did not change significantly [39]. Dautrebande M et al. [17] used the same experimental conditions to stimulate the sciatic nerves of rats in vivo, and the experimental results showed substantial variability because the recorded compound muscle action potential (CMAP) was an order of magnitude larger than the CNAP. To investigate whether INM can cause tissue damage, Wells et al. [40] used NIR with a wavelength of 2120 nm and an intensity of 0.34–0.48 J/cm^2^ to stimulate the rat sciatic nerve, and nerve tissue was isolated immediately and two weeks after stimulation. Histological observations showed no obvious injuries, and the injury threshold (0.66–0.70 J/cm^2^) was much higher than the stimulation intensity. Therefore, IR irradiation of the sciatic nerve is a safe stimulation method.

Other studies have found that IR irradiation can be used to activate the pudendal nerve under certain conditions. The rat cavernous body after radical prostatectomy was stimulated by NIR with a wavelength of 1870 nm and an energy of 0.03–0.67 J/cm^2^. The action potential of the cavernous nerve was recorded, the pressure reflex of the rat penile cavernous body was successfully induced, and the pressure returned to the normal range after stimulation. As the thickness of the cavernous fascia increased, greater stimulation power was required to activate the cavernous nerve [41,42].

### 3.2. Nerve Inhibition with IR Light

At present, the thermal effect of IR is thought to be the main mechanism of neural inhibition. Xia Q. et al. [43] cultured rat cortical neurons on microelectrode arrays to establish a neural network with spontaneous neural activity. Irradiation was performed with 1550 nm NIR at a maximum power density of 56 mW. The effect of INM on neural activity was evaluated by recording the spike emission rate of the neural network. In addition, a temperature model was developed to predict the temperature change under IR radiation at different intensities. INM could inhibit spontaneous neural activity and induce epileptiform discharges within a safe temperature range. In addition, the dose-response relationship between the IR intensity and the inhibitory effect of the neural network was studied. The degree of neural network inhibition increased with increasing IR intensity and temperature.

INM could inhibit pyramidal neurons in the cortex or excite inhibitory neurons, thereby inhibiting the electrical activity of pyramidal neurons. Feng H.J. et al. [44] used a wavelength of 1.85 μm. The amplitude and frequency of spontaneous inhibitory postsynaptic currents (sEPSCs) of rat cortical neurons increased with 0.36 J/cm^2^ NIR radiation, while the decay time constant of the sEPSCs decreased. γ-aminobutyric acid (GABA) could mediate 30–40% of the inhibition of the nervous system.

### 3.3. IR Light Can Be Used to Treat Nervous System Disorders

Several studies have found that INM could treat the symptoms of nervous system disorders (Figure 1). We searched the relevant literature on INM for the treatment of nervous system disorders, which suggests that it can effectively improve cognitive function, reduce pathological damage (Table 3). Because of its noninvasive nature and high spatial resolution, INM is expected to become an effective tool for the treatment of nervous system disorders.

#### 3.3.1. IR Light for Treating Alzheimer’s Disease

Alzheimer’s disease (AD) is a severe neurodegenerative disorder. The associated pathological changes are amyloid plaques and tau protein hyperphosphorylation in brain tissue, and the main symptom is impaired memory function [60]. Nizamutdinov D et al. [52] used a low-power NIR helmet device with an emission wavelength of 1060–1080 nm and a power density of 23.1 mW/cm^2^. After 8 weeks of treatment, cognitive function, sleep quality and anxiety symptoms of AD patients were improved. Other studies found that 8–10 μm FIR at 1 W/cm^2^ could significantly increase epidermal growth factor (EGF) and brain-derived neurotrophic factor (BDNF) levels in 5×FAD mice (a transgenic model of AD); however, their motor and cognitive functions were not improved [58].

In vitro studies have also confirmed the therapeutic effects of IR in AD. Kawasaki, T. et al. [57] applied MIR with a power density of 35–45 mJ/cm^2^ and wavelengths of 5.00 μm, 6.17 μm and 7.19 μm to irradiate PS, APP and tau AD transgenic mouse brain slices, and the results showed that 5.0 μm MIR could dissociate Aβ in brain slices of AD model mice. In contrast, 6.17 μm MIR could dissociate Aβ only in the presence of water. When the slices were irradiated without water, no obvious changes in Aβ structure were observed, indicating that MIR induces the dissociation of Aβ in brain slices but requires the participation of water molecules.

#### 3.3.2. IR Light for Treating Parkinson’s Disease

Parkinson’s disease is a chronic disorder of the central nervous system caused by the slow degeneration of dopaminergic neurons in the substantia nigra compacta. The most commonly used models of Parkinson’s disease are transgenic mice and 1-methyl-4-phenyl-1,2,3,6-tetrahydropyridine (MPTP) compound-induced mice [61]. Reinhart F et al. [48] used 810 nm NIR to irradiate MPTP-treated Parkinson’s disease model mice. The mice were irradiated with NIR light 4 times for 90 s each over a 30 h period, and histological and behavioral experiments were performed at 8 d after the first exposure. The number of dopaminergic nerves and motor activity in the irradiated group were significantly higher than those in the control group. Thus, NIR had neuroprotective effects and improved the athletic ability of Parkinson’s disease model mice.

#### 3.3.3. IR Light for Treating Depression

Depression is a common mental disorder with symptoms such as sleep or appetite disturbance, lack of energy, inattention and negative emotions. After exposure to a series of chronic unpredictable stressors, animals develop symptoms of depression. Wu et al. [49] found that NIR (810 nm, 100 Hz, 20% duty cycle, 120 J/cm^2^, 3 times per week, 3 weeks) could improve the rats’ performance in the forced swimming test. Mohammed et al. [47] used 804 nm NIR with different powers (80, 200 and 400 mW, continuous wave) to continuously irradiate depressed rats induced by chronic reserpine application for one week. They found that IR with 80 mW power had the most significant effects on improving the behavioral of the depressed rats. Electroencephalography (EEG) results showed that 80 mW IR could restore the abnormal EEG activity of depressed rats to normal levels.

#### 3.3.4. IR Light for Treating Traumatic Brain Injury

Traumatic brain injury (TBI) is caused by a disturbance in cerebral blood flow resulting from a severe traumatic injury to the skull. Patients with moderate-to-severe TBI often develop a variety of long-term symptoms after recovering from head injury, such as cognitive impairment (poor memory, impaired executive function, and lack of concentration), headaches, sleep disturbance, and depression. Previous studies have found that IR stimulation can improve cognitive function in TBI patients, showing positive effects on learning and memory and reducing inflammation and cell death in the brain [50]. Khuman et al. [46] used NIR with a wavelength of 800 nm and an intensity of 60 J/cm^2^ to stimulate the injured brain tissue of mice 60–80 min after TBI. Compared with the control group, the average escape latency in the Morris water maze significantly decreased in the irradiated group, and the number of microglia and the level of inflammation were reduced in the irradiated group.

#### 3.3.5. IR Light for Alleviating Drug-Induced Mental Disorders

Methamphetamine (MA) is a highly addictive psychoactive drug, and long-term use will lead to cognitive dysfunction. There are currently no effective medications available for treating MA-induced cognitive dysfunction [62]. Mai et al. [59] constructed drug addiction mice models by inducing protein kinase Cδ knockout (PKCδ KO) and wild-type (WT) mouse with MA, and the memory function of mice was evaluated through a new object recognition experiment. The memory function of PKCδ KO mice was not damaged, while the memory function of WT mice was damaged. After exposure to MIR at a wavelength of 5–20 μm, the memory function of WT mice returned to normal. Western blot results showed that MA promoted PKCδ phosphorylation in prefrontal lobe-induced memory impairment. The level of phosphorylated PKCδ returned to normal after MIR exposure. Therefore, MIR plays a role in alleviating MA-induced memory impairment by regulating PKCδ activity. Other studies have found that MIR could reduce the memory impairment caused by MA by positively regulating the interaction between M1 mAChR and ERK 1/2 [63].

### 3.4. IR Light Can Enhance the Nervous Function of Healthy Individuals

INM can be used not only to treat nervous system disorders but also to modulate nervous function in normal people or healthy animals. Previous research has shown that INM can increase the cerebral blood flow of subjects and improve their emotional and cognitive functions.

To study the effect of IR on the brain network during a spatial memory task, Gutierrez-Menendez et al. [53] irradiated rats with NIR at a wavelength of 1064 nm and an intensity of 20 J/cm^2^ during the Morris water maze reversal task. The results showed that IR reduced the level of cytochrome c oxidase (CCO) in some brain regions involved in the spatial memory task and improved the efficiency of the brain network involved in the task.

In addition, INM had a positive effect on nervous system function in normal subjects. El Khoury H et al. [51] investigated the effects of NIR with a wavelength of 0.81 μm and an average power density of 100 mW/cm^2^ on brain activity at rest and during task performance (finger tapping) using functional magnetic resonance imaging. The results showed that NIR had no effect on cerebral blood flow and the general state of the brain at rest. However, NIR could reduce the activities of the putamen, primary somatosensory cortex and parietal cortex during the task, inhibiting the functions of these regions. Therefore, IR may affect brain activity only in the context of specific tasks.

The psychomotor vigilance task (PVT), the delayed matching-to-sample (DMS) task and the Wisconsin card sorting test (WCST) are reliable indicators of higher prefrontal neural function and are used to assess subjects’ attention, working memory and executive function, respectively [64,65,66]. Barrett D W et al. [54] used continuous NIR with a wavelength of 1064 nm and a power density of 250 mW/cm^2^ to irradiate subjects’ foreheads for 2 weeks, and the control group received 1/12 of the cumulative irradiation dose of the treatment group. After irradiation, the performance of the subjects in the PVT and DMS tasks improved, and the results on the emotion scale indicated a positive emotional state. Blanco N J et al. [55] used the same experimental conditions and found that the irradiation group made fewer errors and showed better performance on the WCST than the control group. Michalikova S et al. [56] exposed female CD1 mice to NIR at 1072 nm, and the irradiated mice performed better in the 3D maze test than the control mice. Salgado A S et al. [45] stimulated the occipital lobe, temporal lobe and parietal lobe of healthy subjects with NIR at 903 nm. After 2–4 months, quantitative EEG results showed that the amplitude of the subjects’ EEG activity increased by an average of 0.81 μV.

## 4. Mechanism of IR Light in Regulating Neural Function

At present, it is generally accepted that the photothermal effect is the main mechanism by which IR regulates neural function. IR irradiation can be absorbed by water, generating heat. Temperature changes can activate photosensitive ion channels, create transmembrane capacitors or temporary nanopores, and thereby affect the electrical activity of neurons [67]. However, excessive photothermal effects can cause cell and tissue damage. In addition, IR irradiation can affect energy metabolism through nonthermal effects [68].

### 4.1. Activation of Thermosensitive Ion Channels

The conductance of ion channels is affected by temperature to some extent, and INM may directly activate thermosensitive ion channels in the nerves [69]. A schematic of this process is shown in Figure 2. The transient receptor potential vanilloid (TRPV) family is the major thermosensitive ion channel in the nervous system, which can be used as a receptor for capsaicin and vanilla compounds and can be activated under acidic or hot conditions. According to the thermal threshold reported for HEK293 cells and Xenopus oocytes, the activation sequence of TRPV channels is as follows: TRPV4 > 27 °C, TRPV3 > 32 °C, TRPV1 > 43 °C, and TRPV2 > 53 °C [70,71]. Barrett et al. [72] used 1863 nm NIR to stimulate the spiral ganglion neurons of mice. In the absence of Ca^2+^ in the culture medium, the neurons could generate action potentials via Ca^2+^ released by TRPV4 of the endoplasmic reticulum. Auditory brainstem responses (ABRs) are reliable indicators of cochlear function. Suh et al. [73] irradiated the cochlea of TRPV1 gene knockout mice with 1.85 μm and 1.86 μm NIR at a power density of 0–1 J/cm^2^. Compared with the control group, the cochlear nerves of TRPV1 gene knockout mice did not generate action potentials during the exposure period, and the threshold for ABR induction was significantly increased. The results showed that TRPV1 is involved in the generation of IR-induced action potentials.

### 4.2. Increased Membrane Capacitance

Shapiro et al. [74] investigated the effects of NIR at wavelengths of 1889 nm and 1869 nm and energies of 0.28–7.3 mJ on the transmembrane potential of oocytes, HEK293 cells and artificial lipid bilayers through voltage clamp and current clamp experiments. After blocking the ion channel, IR could still increase the cell capacitance, and the increased capacitance was positively related to the IR dose, thus reversibly inducing the action potential. The change in the membrane capacitance depended on the time gradient of the temperature increase rather than the absolute temperature. The increased membrane capacitance could persist for 100–200 ms after the application of a single stimulus. Plaksin et al. [75] found that the membrane capacitance changed by approximately 0.3% for each 1 °C increase through mathematical model analysis. In addition, previous studies on the neuromuscular junction of *Caenorhabditis elegans* have shown that the rapid temperature increase caused by pulsed IR irradiation can lead to a redistribution of membrane surface charges, and the resulting asymmetric excitation current leads to the depolarization of the cell membrane [76]. Therefore, the increase in membrane capacitance caused by the temperature increase may also be an important mechanism by which INM regulates neural activity [77].

### 4.3. Instantaneous Nanopore Formation on the Cell Membrane

In addition to activating photosensitive ion channels and increasing membrane capacitance, the temperature gradient generated by IR irradiation can increase the instability of phospholipid components in the cell membrane, leading to the formation of transient nanopores, which allow extracellular ions to enter the cytoplasm.

Beier et al. [78] exposed rat primary hippocampal neurons, NG108, CHO-K1 and other cells to NIR at a wavelength of 1890 nm and an energy of 0.6–5.1 mJ. After blocking the ion channel, Ca^2+^ influx was observed, cell swelling volume increased, cell membrane permeability increased, and action potentials were induced. Moreover, the influx of Ca^2+^ triggered a series of cascade reactions, activating the phosphatidylinositol lipid signaling pathway and leading to the generation of the second messenger inositol triphosphate (IP3) and diacylglycerol (DAG) in cells. IP3 binds to IP3 receptors in the endoplasmic reticulum, inducing the release of Ca^2+^ and leading to a large influx of extracellular Ca^2+^. In addition, the intracellular Ca^2+^ activated protein kinase C (PKC). PKC was attracted by the effective DAG and translocated to the proximal membrane space to perform its physiological functions. The activation of this pathway may also play a role in the observed swelling and blistering. NG108 cells are undifferentiated cells derived from brain tumors that cannot generate action potentials, and a gradual increase in Ca^2+^ was also observed under these conditions, suggesting that this phenomenon might be mediated by the formation of ion channel-independent cell membrane pores. The influx of fluorescent markers demonstrated the change in the cell membrane structure. Therefore, IR stimulation could cause Ca^2+^ to flow into the cell by changing the cell membrane structure, thereby activating the intracellular signaling pathway.

### 4.4. Stimulating the Mitochondrial Respiratory Chain

CCO is a vital component in the mitochondrial respiratory chain. The electrons generated by glucose metabolism ultimately reduce oxygen to water. Nitric oxide (NO) inhibits electron transfer in the respiratory chain. CCO enhances enzyme activity by absorbing photon energy at specific wavelengths to dissociate NO. This leads to an increase in the mitochondrial membrane potential (MMP), an increase in oxygen consumption, and increased ATP production by mitochondria, improving neuronal respiration and metabolism. In addition, photolytic NO dilates blood vessels, increases blood flow, and improves oxygen supply to the brain [79,80]. When CCO absorbs photons, reactive oxygen species (ROS) and Ca^2+^ in mitochondria can be transiently increased, which could trigger numerous mitochondrial signaling pathways and produce cell-protective, antioxidant and anti-apoptotic effects. Excessive ROS activate the translocation of the transcription factor NF-κB to the nucleus, inducing the expression of proinflammatory cytokines and nerve inflammation [81]. A schematic of the stimulating effect of IR light on the mitochondrial respiratory chain is shown in Figure 3.

Xu et al. [82] irradiated mice with 808 nm, 23 mW/cm^2^ NIR radiation. In this case, the head of the animals was shaved, covered in foil with a window of a certain size and location, and the fiber optics ended close to this window. After NIR radiation, CCO activity in the hippocampus and prefrontal lobe of depression model mice was significantly increased, and abnormal mitochondrial metabolism was ameliorated. The results of the forced swimming test and tail suspension test showed that the depression symptoms of the mice were alleviated. Tian, F. et al. [83] found that brain oxygenation and hemodynamic parameters were improved during and after NIR irradiation with a wavelength of 1064 nm and an average power density of 0.25 W/cm^2^.

Therefore, IR irradiation can improve metabolic levels, dilate blood vessels, increase cerebral blood flow, and regulate neurobehavioral activities through the mitochondrial respiratory chain.

## 5. Outlook

This article reviews the characteristics of infrared light, the advantages and disadvantages of infrared neuromodulation, its role in improving personal health and its mechanism.

Although the development of INM methods is in the early stage, INM shows potential as a neuroscience tool. Many studies have found that IR irradiation has certain positive effects on disorder treatment and cognitive improvement under various conditions, and its mechanism is gradually being clarified. At present, most studies have focused on the NIR band. With increasing research, the neural function regulation of MIR and FIR has attracted considerable attention. In recent years, the terahertz wave field has developed rapidly and has been widely used in communication, navigation, imaging, security inspection and other applications. Biomedical applications and THz wave effects are currently being studied. Safety is the main issue in the application of INM to the human body. The safety of INM in the peripheral and central nervous systems has been verified. The temperature changes observed during INM and optogenetic stimulation are similar. This article reviews the characteristics of infrared light, the advantages and disadvantages of infrared neuromodulation, its effects on improving individual health, and its mechanism (Figure 4). The specific parameters of INM application, such as the wavelength, energy density, power density, number of repetitions, and optical transmission mode, need to be investigated further. With the development of optical technology and equipment, laser or LED devices will be more widely used, and INM will become an important tool for regulating neural function.

## Figures and Tables

**Figure 1 ijms-25-00928-f001:**
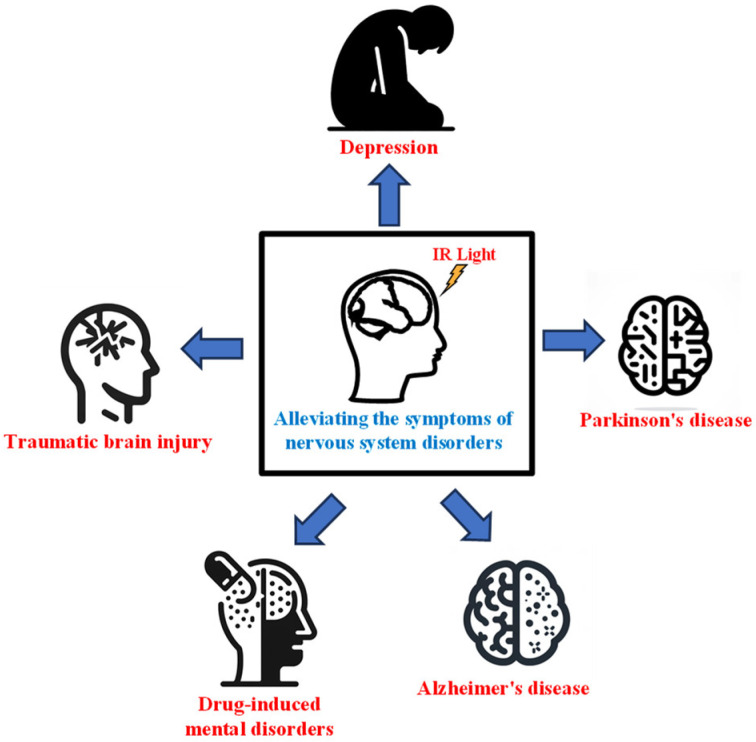
IR light could improve the symptoms of nervous system disorders.

**Figure 2 ijms-25-00928-f002:**
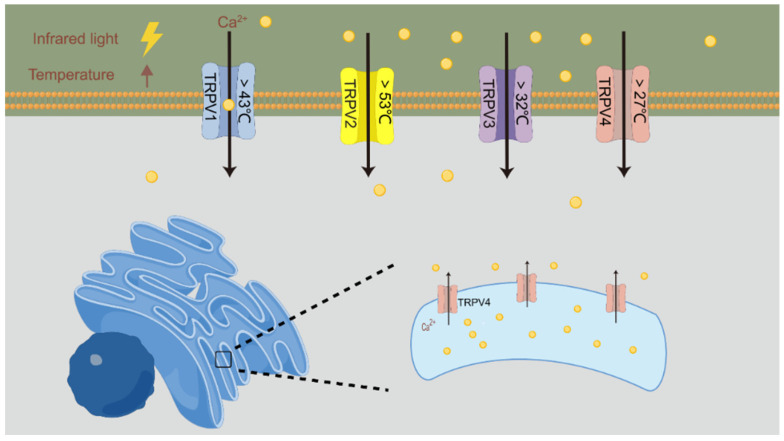
IR light activates TRPV channels by increasing the temperature.

**Figure 3 ijms-25-00928-f003:**
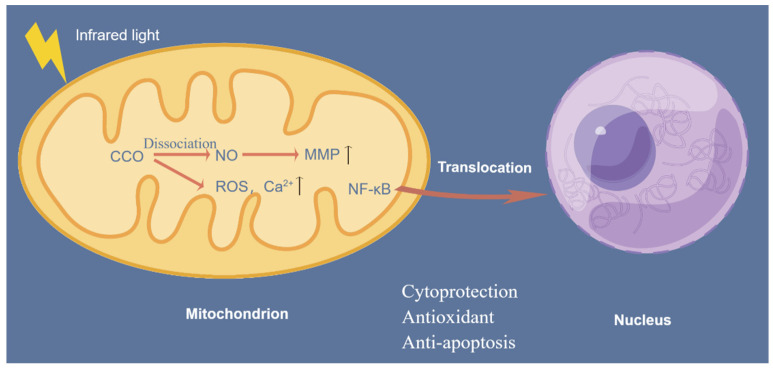
The effect of IR light on the mitochondrial respiratory chain.

**Figure 4 ijms-25-00928-f004:**
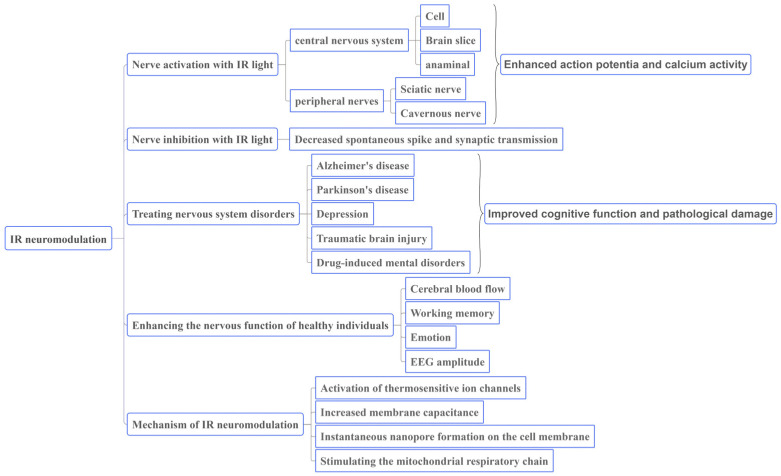
The overall structure of research on IR light regulation of neural function.

**Table 1 ijms-25-00928-t001:** Classification of IR light.

IR Category	Wavelength (μm)	Frequency (THz)
NIR	0.75–2.5	120–400
MIR	2.5–25	12–120
FIR	25–1000	0.3–12

**Table 2 ijms-25-00928-t002:** Comparison of different neuromodulation methods.

Neural Modulatory Methods	Applications	Advantages	Disadvantages	References
Electrical stimulation	Treating disorders of the nervous, cardiovascular, and sensory systems	Mature technology and wide applicability	Poor spatial specificity, susceptibility to environmental electrical interference, and risk of tissue trauma	[15,16,17]
Magnetic stimulation	Noninvasive treatment of central and peripheral nervous system disorders	Safe, noninvasive, and highly tissue penetrability	Inaccurate spatial locations and individual variations	[18,19]
Optogenetics	Precise modulation of neuronal subtypes at temporal and spatial scales	Precise modulation with high temporal and spatial resolution	Requires invasive surgery with fibre implantation	[20,21,22]
Visual stimulation	40 Hz flicker stimulation reduces Aβ and phosphorylated tau levels in AD transgenic mice; 3000 lux of bright light has antinociceptive effects on mice	Noninvasive and less adverse reactions	Cannot target specific brain regions, and the mechanism remains unclear	[23,24]

**Table 3 ijms-25-00928-t003:** Effects of different IR light conditions on neurological disorders.

Experimental Model	Exposure Conditions	Effects	Reference
Healthy elderly women	627 nm, 70 mW/cm^2^, 2 min/d, 4 d/week, 4 weeks	Blood and vasomotor behaviors of the basilar and middle cerebral arteries were promoted	[45]
TBI mice	800 nm, 60 J/cm^2^ 60–80 min	Memory function was restored and inflammation was reduced	[46]
Reserpine-induced depression rats	804 nm 80, 200 and 400 mW, continuous wave 6 min/d, 7 d	After 80 mW IR exposure, the behavior of depressed rats improved, and the EEG activity returned to normal	[47]
MPTP-induced Parkinson’s disease mice	810 nm, 160 μW	Number and motor activity of dopaminergic nerves increased	[48]
Chronic mild stress-induced Wistar rats	810 nm, 100 Hz, 20% duty ratio, 120 J/cm^2^, 3 weeks	Forced swimming performance improved	[49]
TBI patients	810 nm and 980 nm, 10–15 weeks	Neurological function improved, and cerebral inflammation decreased	[50]
Human subjects	810 nm, 100 mW/cm^2^ 20 min	Function active task-related brain regions were suppressed	[51]
AD patients	1060–1080 nm, 23.1 mW/cm^2^, 8 weeks	Cognitive function and sleep quality improved, anxiety symptoms decreased	[52]
Wistar rats	1064 nm, 30 mW 1 h/d, 5 d	The activity of the brain network involved in the spatial memory task was more efficient	[53]
Human subjects	1064 nm, 250 mW/cm^2^, 2 weeks	Cognitive, emotional and executive functions in humans were improved	[54,55]
CD1 female mice	1072 nm 6 min/d, 10 d	Performance in 3D spatial navigation task improved	[56]
APP/PS1/Tau transgenic mouse brain slices	5.00, 6.17 and 7.19 μm, 35–4 5 mJ/cm^2^	Aβ fibrils were dissociated	[57]
5 × FAD mice	8–10 μm, 1 W/cm^2^, 5 months	EGF and BDNF levels increased significantly	[58]
MA-induced mental disorder model	5–20 μm 14 d, 40 min/d	Memory function was restored	[59]

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
