# Peer review of "Advances in the Regulation of Neural Function by Infrared Light"

_ijms, 2024, doi:10.3390/ijms25020928_

Round 1
Reviewer 1 Report
Comments and Suggestions for Authors
This literature review sums up literature on infrared light as modulator for the nervous system.
It is well written and well organized. As far as I can tell, it contains the majour publications on this topic. Overall I found it clear, but in several cases had trouble understanding the experiments listed. For example, it is often unclear whether the infrared light was applied to the intact animal, and how, or to cells or tissues of the animals. Below I have listed which paragraphs I suggest changing in this respect, but please work through the whole review to clarify how the experiments were performed. Second, there is a chapter that discusses whether the proven effects of infrared light are caused by heat or by the light itself. This is an important topic and as far as I understand this review, in many cases it was proven that the actual effect was by the heat produced by the infrared rays. However this review remains a bit vague whether there are any cases in which it was proven that light (not heat) causes the effects. Please specify.
Overall I recommend publication once the points listed below have been addressed.
Lines 62 ff: When I read the definition of “neuromodulation” I found two important types of neuromodulation missing – first of all many times neuromodulation is performed with drugs, which should also be listed in line 70. But perhaps more importantly, the given definition only covers the technological part of neuromodulation (humans interfering with the nervous system). Neuromodulation also occurs naturally and neuromodulators are defined as substances that have an influence on neurons and usually act more slowly than neurotransmitters. Please include this definition for clarity.
lines 104/105: “NIR activation mainly leads to thermal effects, while MIR can activate the central nervous system under nonthermal conditions.” There is no citation, so please quote the right citation. Or remove here since there are more paragraphs on this topic below.
ll- 106 – 117: unclear whether the experiments were done invasively or not, please specify.
l. 109: “tactile stimulation” – what was stimulated and how?
ll. 148-168: this paragraph is about NIR, whereby the chapter starts with NIR, then turns the MIR, and now back to NIR. Please move this paragraph up to line 126 to be more consistent. Or, if you have a reason for keeping the current order, please make the reason clear.
ll. 234 ff: time scale of treatment lacking
l. 240 “number of dopaminergic nerves”. Surely the nerve number did not change nor was it counted. Please specify what exactly changed.
ll. 272-275: confusing sentence, please clarify.
l. 328, this chapter talks about “photosensitive” channels but when reading carefully, the channels seem to be temperature dependent, not photo sensitive. Please clarify!
l. 384 “CCO” please spell out this abbreviation as it is mentioned here for the first time.
l. 411- 418: Explain the stimulation protocol of the experiment in more detail please- in this case the head of the animals was shaved, covered in foil with a window of a certain size and location, and the fibre optics ended close to this window.
Reviewer 2 Report
Comments and Suggestions for Authors
The paper serves a topical review on the use of IR radiation on neural regulation. The topic is known but summing the state of the art is a fruitful idea, and such a review has a practical value for researchers of several specialties.
Some relatively minor issues should be taken into account^
1. The starting part (the beginning of the abstract), the starting part of the Introduction have phrases that are truisms: "Infrared light is part of the electromagnetic spectrum. It is widely present in the environment. " and many more.
Such phrases should be replaces with something more scientifically sound or removed completely.
2. For such a full-cover review the methodology of reference search should be added to the Introduction: year range, databases, keyword, and search phrases.
3. Using 'infrared' after adding the obvious and common abbreviation of 'IR" seems to abundant, it makes many discussion points too wordy.
4. The reference style is not IJMS's, it should be corrected.
5. Table 3 is not well organized: the order of references is too chaotic. Fro the logic of discussion, it should be sorted by the wavelength. Another fruitful sorting is the energy/fluence/dose. Right now the value of this summary table seems degraded.
6. I recommend adding some figures, in such a topic they should be quite illustrative.
7. Probably, in concluding remarks (or in the main discussion) an authors' figure with te structure of the whole research could be quite useful. The authors discuss it in the text, but the whole drawn picture involving the tasks, objects, Ir ranges and the intensities or doses may be quite useful.
Comments on the Quality of English LanguageI believe that the text should be double-checked for modal verb use ('may/can/might/could' etc.) and singular/plural use, and similar grammar issues, the text contains rather many misuses.
